# The Visual Behaviour of the Cyclist: Comparison between Simulated and Real Scenarios

**Ennia Mariapaola Acerra** [1,*], **Murad Shoman** [2], **Hocine Imine** [2], **Claudia Brasile** [1], **Claudio Lantieri** [1] **and Valeria Vignali** [1]

1 Department of Civil, Chemical, Environmental and Material Engineering (DICAM), University of Bologna, Viale Risorgimento 2, 40126 Bologna, Italy
2 Laboratoire Perceptions, Interactions, Comportements Simulations des Usagers de la Route et de la Rue (PICS-L), Components and Systems Department (COSYS), Gustave Eiffel University, 77420 Champs sur Marne, France
* Correspondence: ennia.acerra2@unibo.it

**Abstract:** Cyclists are one of the main categories of road users particularly exposed to accident risk. The increasing use of this ecological means of transport requires a specific assessment of cyclist safety in terms of traffic flow and human factors. In this study, a particular visual tracking tool has been used to highlight not only the main critical points of the infrastructure, where a high level of distraction is recorded, but also the various interactions with different road users (pedestrians, vehicles, buses, wheelchairs, cyclists). To confirm the critical aspects of the infrastructure and the trend of workload, a similar circuit was reproduced in a bicycle simulator, which also allowed a meaningful comparison of cycling behaviour. The innovative component of this paper is a comparison between a real test, held in Stockholm, and a simulator where the same scenario has been represented, in order to highlight the objective differences in behaviour. The cycling performance was also evaluated both from an objective point of view, with the count of frames related to each category of visualization, and from a subjective one, through the questionnaires. The results show the crossing as a critical aspect because only 4/3% fixation is required for both simulated and real tests to confirm the significance of the comparison between the two experiments. The high attention rate, resulting from frame-by-frame analysis, also points to a clear difference in the perception of users, who feel with a low workload.

**Keywords:** visual behaviour; bicycle simulator; eye tracking; cyclist safety

## 1. Introduction

The use of bicycles appears to be strongly growing in recent years, due to an increasing awareness of the environmental impact of transport, and strategies for reducing harmful emissions. The behaviour of cyclists, therefore, is becoming one of the main elements to be investigated for road safety, as they present an increasing percentage of road users. The main methodology for investigating the cyclist's behaviour and the elements that influence or distract them, based on how the cyclist chooses to act in one way rather than another, is through the study of their visual behaviour. This paper studies the behaviour of cyclists through highly innovative eye-tracking instrumentation, which allows attention during the test and the performance of road manoeuvres to be verified. The innovative component of the paper is a comparison between a real cycle ride in Stockholm, and a simulation of the same scenarios, highlighting the objective differences in behaviour. The use of the Pupil Core eye-tracking system allowed the number of frames dedicated to each element encountered on the road to be quantified and the points of possible danger or interference to be defined, dictated by a high inattention of the cyclist.

*1.1. Eye Tracking Applied to Road Safety*

Studying visual behaviour means evaluating the sequence of interactions, called 'visual events', between the observer and who or what is being observed. By observing gaze movements and, through them, analysing how the individual is able to reach certain levels of attention. Indeed, it is possible to define visual behaviour in relation to specific actions or scenarios that come to be determined in the external world [1–4]. This methodology provides a clear view of a user's perception while they are performing a certain action, through feedback of a subjective nature, but without objectively describing the problems experienced during the action. For this reason, it is necessary to use eye-tracking technology, which allows an objective calculation of the mechanisms of human vision used in different fields to be obtained, such as neuromarketing, literacy processes, psychology, medicine, and driving behaviour [5,6]. It permits the most relevant visual events to be highlighted, considering what and how long a subject is observing, in addition to recording the contraction of pupils, which are clear signs of cognitive input for the variation of the workload. In particular, this technology studies visual behaviour to understand cognitive and emotional processes, providing theoretical and conceptual approaches. One of the main advantages of this technique is themanageability of the instrument, i.e., innovative glasses, which not only allow the acquisition of information on the view but also provide data on brain function continuously. However, an objective evaluation of the point of view, extrapolated by an eye-tracking system, does not exclude a psychological evaluation, just as important as the subjective perception [7–10]. The use of questionnaires or interviews is of assistance to acquire information regarding the perception, the workload, and the effort of the individual to perform a certain action. The questionnaires allow us to extrapolate first the behaviour and then the comprehensive psychological framework [11–13].

Today, eye tracking is an analysis method that is well-developed in the field of mobility. The view represents the source of 90% of the information required to drive; organizing and deciphering the data from the external environment allows the establishment of the basic parameters for safe driving. The eyes are the most stimulated and stressed organs while driving, as they have the task of collecting primary stimuli from the controls of the car, the management of road warnings, and interactions with other road users [12]. In addition, the road user modulates their behaviour by considering not only their habits but also external factors. Therefore, it is essential to study the trend of the gaze, through an eye-tracking system, to define useful parameters for road safety [14]. One example is the factor of attention and, consequently, distraction. Visual attention imparts awareness of the outside environment, and it contrasts with the concept of distraction, which interferes with driving performance [15]. Driver distraction is defined as a variation of attention, followed by temporary concentration on non-driving-related actions. This results in a reduction in performance quality, causing possible risk situations [16,17]. Therefore, driver distraction is caused by the performance of secondary activities that take the eyes off the main job [14,18,19]. When the user manages primary and secondary tasks simultaneously, an important factor becomes relevant: the driving experience. According to Crundall [20], experienced drivers are able to capture visual strategies that depend on the complexity of manoeuvres and alignment, whereas less-experienced drivers process a lower amount of information that leads them into more dangerous situations [21]. Among the main secondary tasks responsible for inattention driving is the use of a mobile phone [22]. Many researchers underline that mobile phones affect performance negatively. In fact, the visual–manual activities compromise the duration of the gaze on the area of interest, reducing it considerably [23,24]. For users, there are two important aspects during the driving action: their psychology, with the perception of the outside world; and their behaviour toward road users. Therefore, it is necessary to understand which elements are most influential while driving, considering attention and inattention, and which can compromise the level of road safety, to carry out an analysis with an eye tracker tool. Crundall [25] studies the percentage of time spent observing the surrounding scenario; which is about 20–50% of the total time, thus highlighting more viewing of distracting items. Numerous studies have

examined the physical elements of the road that may obstruct the driver's view or vehicles leaving the road [26]. As part of Human Factors, one of the crucial elements to consider is the study of eye-catching objects, that is the elements present in the road layout that could modulate the driver's attention, according to their positioning. In the bibliography, one of the analysed objects is represented by the billboard. Indeed, considering the position, symmetrical or asymmetric, the path, or the impact of colour, this object could represent a possible distraction factor for the driver, leading-to high-risk situations [27,28]. The lack of clarity of the route is the second aspect that can compromise road safety, concerning an inconsistent design of infrastructure away from the concept of 'self-explanatory roads'. This type of road, defined as user-friendly, allows the identification of possible critical points with an appropriate advance for speed modulation [29–33].

*1.2. The Visual Behavior of Road Users*

The analysis of visual behaviour is also useful for observing the mutual relations between road users [34]. Sometimes, the driver's behaviour and level of attention translate into a 'black event', which happens when the driver does not perceive other road users as a real danger, or when a user makes incorrect considerations about the user's future actions [35]. This type of event is particularly frequent when vehicles interact with bicycles. Cyclists, are particularly vulnerable users, most exposed to the accident risk factor for several reasons [34,36–38]. First of all, a cyclist's field of vision is far wider than a driver's car [34,39,40]. In addition, the cyclist is a 'direct victim' of weather conditions, that could compromise visibility and balance, if the road surface features are considered [30]. However, when vehicles interact with the infrastructure, the performance of riders are adversely affected [34]. This article illustrates the visual behaviour of the driver with these two important aspects. In many cities, the use of bicycles is becoming more widespread, highlighting several positive effects in terms of environmental sustainability. Therefore, it is useful to deepen the aspects that could affect the performance of the cyclist, to achieve a future where cities are cycling-friendly [29]. From this perspective, the actual experiment allowed us to define the visual and driving behaviour of cyclist objectively, exploiting an innovative tool that is the Mobile Eye Tracker. In particular, the behavioural data from a bicycle simulator compared with behavioural data from the site represents the innovation of this research. Simulators are useful to assess how the user lends themself to certain issues, such as learning to drive, testing new road features, and conducting road safety investigations [41,42]. The main advantage of bicycle simulators is the possibility to create different situations and especially the desired conditions for research and avoid the risks associated with a real environment [43]. To determine the most effective comparison, a scenario was introduced with the same characteristics as the real one, located in Stockholm. The use of the PICS-L bicycle simulator allows the circuit to be reproduced with functional and mechanical features. It is one of the most effective simulators in the world that, for example, differs from the KAIST interactive bicycle simulator, as it provides not only the scenery but also simulates vibrations and skids that typically occur on the road [22,30,44–52]. The results have led to important evaluations that are excellent cues both to evaluate the critical points of the infrastructure and to elaborate the levels of attention that depend on the type of road.

## 2. Materials and Methods

*2.1. Experimental Procedure*

In total, 40 users were recruited for testing. None of the participants wore glasses or lenses to obtain a homogeneous sample, which could avoid possible artifacts in eye-movement monitoring. A total of 20 of them were engaged for the on-site test (Mean age = 35.15; SD = ±13.7) and 20 users for the simulator experiment (Mean age = 27.47, SD = ±4.5). All participants rode the same route: one of 4 km located in the north of Stockholm (Sweden) and the other reproduced in the simulator (the simulator route is half

the length of the Stockholm route because of technical limitations, but it contains all the zones and the main important infrastructural elements).

Participants represent a homogeneous and statistically significant sample, composed of 22 males (11 for the on-site test, 11 for the simulator test) and 18 females (9 for the on-site test, 9 for the simulator test). They were recruited through social networks and posters in the universities where the tests were carried out. Participants had an average cycling experience of 30.9 years (SD: ±15.9) for males and 26 years (SD: ±15.8) for females and they used their bicycle every day, to reach their places of work or study. For the on-site test, 64% of the users had familiarity with the experiment route (57% of males, 75% of females). However, none of them had used a bicycle simulator before [53,54]. No user knew the purpose of the test, so as not to affect the results. Before the test, all users were provided with relevant information material: the route to be followed, the experimental procedure, and the instruments used, in terms of the operation and calibration phase.

The circuit is divided into four zones according to the characteristics of the infrastructure and the presence of specific types of users: Zone 1 (A and B) represents a combined cycle and vehicle route, without specific separation signals; Zone 2 comprises a carriageway where part of the road has been designated as a cycle lane, divided by horizontal road markings; Zone 3 is a pedestrian and cyclist-friendly route surrounded by car parks (Figure 1) [55].

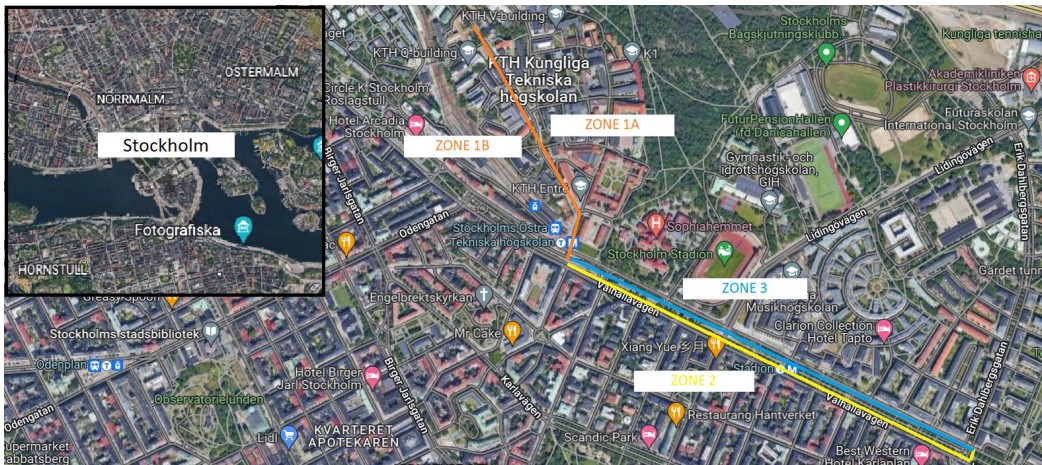

**Figure 1.** Localization of the route and distinction in different zones.

The first trial was on-site. The participants were involved in a road test where the start and finish points coincided with the laboratory of the Royal Institute of Technology in Stockholm (KTH). All participants were asked to sign a standard consent form including brief details about the experiment, the data collected, and the following analysis. They were obliged to wear a helmet and to follow the circuit indicated on the GPS placed on the handlebar of the bicycle, while the simulator test was performed by making a round trip of 2 + 2 km i.e., two laps of the course: the first focuses on adaptation to technologies whereas the second underlines the evaluation of the test. After participants completed the cycling session, they were asked to fill in two questionnaires, to evaluate their subjective perception: the NASA task low index and the disease questionnaire. The first questionnaire consists of six categories of assessment: mental question, physical question, temporal question, performance, effort, and level of frustration (Figure 2).

| Metal demand: | | | | | | | | | | | | | | | | | | | | |
|---|---|---|---|---|---|---|---|---|---|---|---|---|---|---|---|---|---|---|---|---|
| Low | | | | | | | | | | | | | | | | | | | | High |

| Physical demand: | | | | | | | | | | | | | | | | | | | | |
|---|---|---|---|---|---|---|---|---|---|---|---|---|---|---|---|---|---|---|---|---|
| Low | | | | | | | | | | | | | | | | | | | | High |

| Temporal demand: | | | | | | | | | | | | | | | | | | | | |
|---|---|---|---|---|---|---|---|---|---|---|---|---|---|---|---|---|---|---|---|---|
| Low | | | | | | | | | | | | | | | | | | | | High |

| Effort: | | | | | | | | | | | | | | | | | | | | |
|---|---|---|---|---|---|---|---|---|---|---|---|---|---|---|---|---|---|---|---|---|
| Low | | | | | | | | | | | | | | | | | | | | High |

| Performance: | | | | | | | | | | | | | | | | | | | | |
|---|---|---|---|---|---|---|---|---|---|---|---|---|---|---|---|---|---|---|---|---|
| Good | | | | | | | | | | | | | | | | | | | | Poor |

| Frustration: | | | | | | | | | | | | | | | | | | | | |
|---|---|---|---|---|---|---|---|---|---|---|---|---|---|---|---|---|---|---|---|---|
| Low | | | | | | | | | | | | | | | | | | | | High |

**Figure 2.** The NASA TLX questionnaire.

Through the average value, it was possible to derive a subjective assessment of the workload perceived during a test of the scores of each category declared by the participant. It has been shown that the NASA TLX questionnaire is a good alternative to the use of electroencephalography (EEG) and provides the significance of the species values if administered before and after the test [54]. The use of such questionnaires has been fundamental to compare the objective visual data from the eye tracking instrument, and the subjective perception of the user; thus, to estimate the effectiveness of the simulation itself. Before both trials, the eye-tracking instrument was calibrated, and the sickness questionnaire was also done to evaluate fatigue, headache, eyestrain, difficulty focusing, increased salivation, sweating, nausea, difficulty concentrating, the fullness of head, blurred vision, dizziness (eyes open), dizziness (eyes closed), vertigo, stomach awareness, and burping.

*2.2. Instrument and Data Analysis*

All participants wore the Pupil Core for visual monitoring. Pupil Core is an eye-tracking system used to capture the pupil data of the drivers with the available gaze accuracy of $0.60°$ and gaze precision of $0.02$. The glasses consist of two cameras: the 'eye camera', that records the movements of the pupil and the 'scene camera' that collects the frames related to the external environment (Figure 3). The instruments have been calibrated before the experiment for each participant (Figure 4). The eye-tracking calibration provides the parameters to a matrix that correlates the eye movement, from the eye camera with the field of view, with the scene camera.

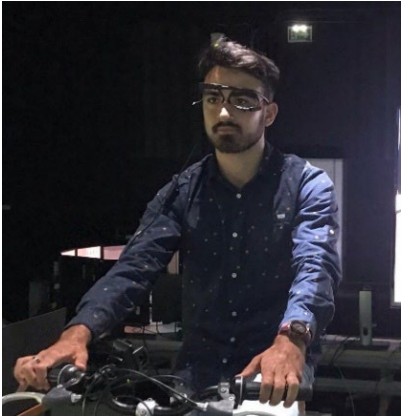

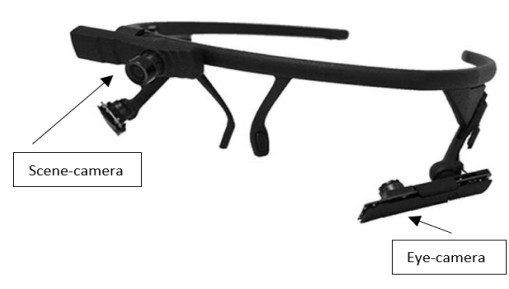

**Figure 3.** Pupil Core.

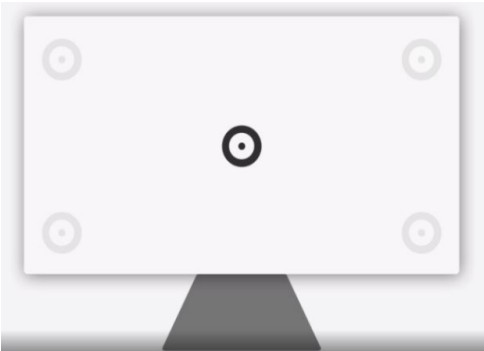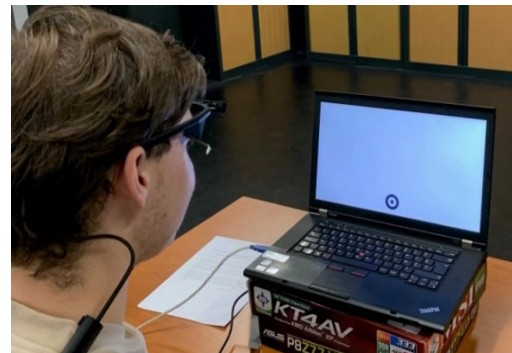

**Figure 4.** Calibration phase.

The 5-point calibration method has been used, which allows the rapid detection of the gaze, using pupil-acquisition software. The subject, without moving the head, must concentrate their look on every red point, localized in the corners and in the centre of the screen; subsequently, when the point becomes green, it proceeds to the verification of the other points. The software repeats the procedure until it reaches an accuracy for the appearance position of 0.60 (Figure 5). After data had been acquired using pupil capture with a laptop, the Pupil Player software was used to post-process the eye-tracking data [53]. Through the overlap of the two images linked to the pupil and the external environment, it is possible to obtain a video that evaluates the ocular path in relation to the sequence of external images, both in the external environment and in the simulator (Figures 5 and 6).

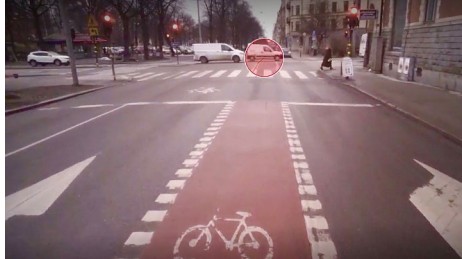

**Figure 5.** The frame of the on-site test.

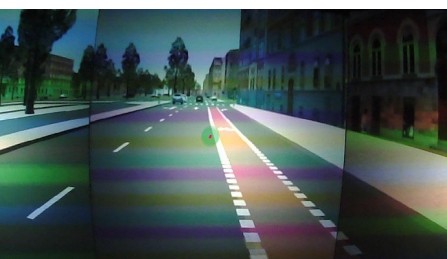

**Figure 6.** The frame of the simulated test.

The Pupil Core video was analyzed frame-by-frame, in order to verify the elements fixed on by each participant. The main categories of analysis are [56–58]:

- Infrastructure, which includes sidewalks and streets;
- Users, correlated with cars, parked cars, pedestrians, and bicycles;
- Signs, considering horizontal, vertical, pedestrian passage, and traffic lights;
- background, including buildings, vegetation, street lamps, and sky;
- Bicycle tests, such as handlebars, pedals, and GPS.

Each group has a defined value related to attention, i.e., infrastructure, signs, and users, or inattention, i.e., background and bicycle test.

## 3. Results and Discussion

### 3.1. On-Site Test

The on-site test shows interesting data in terms of attention. In particular, it has very high and approximately constant percentages for all the areas analysed. In detail, the attention rate of each zone decreases with the progress of the test (variation of a mean of 3%), highlighting that the participants are familiar with tools and road alignment. Indeed, the total duration of fixation begins at 934 sec for the first zone, goes up in zone 2, and begins to decrease from zone 3, then ends with values equal to 1197 sec in the last zone (Table 1). Regarding the trend of the percentages of each user, the behaviours adopted during the test are homogeneous; for each user, the percentage of inattention remains within a very narrow range, from a minimum of 10% to a maximum of 14%.

**Table 1.** Total fixation and duration considering the attention and the inattention.

| Zones | Total Frames | Total Fixation Duration [s] | Fixation Duration of Attention [s] | Percentage of Attention [%] | Fixation Duration of Inattention [s] | Percentage of Attention [%] |
|---|---|---|---|---|---|---|
| 1A | 23,359 | 934.36 | 845.32 | 90 | 89.04 | 10 |
| 2 | 50,545 | 2021.8 | 1756.34 | 88 | 265.46 | 12 |
| 3 | 44,412 | 1776.48 | 1547.35 | 87 | 229.13 | 13 |
| 1B | 29,924 | 1196.96 | 1032.03 | 86 | 164.93 | 14 |

The highest attention rate recorded in zones 1A and 2 is 90% (SD = 0.083) and 87%, respectively (SD = 0.07). The category most attractive for users is infrastructure, in particular the road. Overall, 78% of the total attention frames are focalized on the infrastructure; this shows that the participants mainly looked at the central area of the pavement to keep track of the route, in order to avoid obstacles and holes and to prevent dangerous interactions with other road users (Table 2). The prevalence of attention shows that users are satisfied with road signs and areas dedicated to them, as they focus on the main task of driving, not being devoted to secondary tasks, such as the possible attractiveness linked to advertising signs, the background, or particular elements of the track. On the other hand, they are well aware of being one of the weakest road users and the most exposed to interactions with vehicles, so they are not relaxed. Not surprisingly, in fact, the greatest attention in zones 1a and 2 are recorded in areas where the lane reserved for bicycles is not physically separated from the lane dedicated to vehicles, highlighting that the cyclists have to always be vigilant of the vehicular flow.

**Table 2.** Categories of attention.

| Categories | Total Frames | Total Fixation Duration [s] | Average Percentage [%] |
|---|---|---|---|
| Sidewalk | 3013 | 121 | 2 |
| Street | 101,635 | 4065 | 78 |
| Car | 5916 | 237 | 5 |
| Parked car | 2818 | 113 | 2 |
| Pedestrian | 5617 | 225 | 4 |
| Bicycle | 1417 | 57 | 1 |
| Horizontal Signs | 803 | 32 | 1 |
| Vertical Signs | 755 | 30 | 1 |
| Pedestrian passage | 2138 | 86 | 2 |
| Traffic light | 5414 | 217 | 4 |

Zone 3 and 1B are road sections with lower attentiveness, with 10% and 14%, respectively. The reason for the low attention rate in Zone 3 is likely related to how road users interact with each other. In fact, in this area, there are only moving pedestrians, who present a high level of danger of collision. As a result, cyclists tend to get distracted by having a

low workload and feel safer as they are separated from vehicle traffic. The 4% decrease in attention in zone 1B compared to 1A, however, is linked to the knowledge of the track, as although it is a combined cycle and vehicle route, has a lower number of vehicles than in zone 2 due to the absence of buses. Finally, to outline a cumulative figure on the attention of cyclists during the entire route, it is possible to identify 88% of attention (SD = 0.58) and 12% of inattention (SD = 0.59).

From the cumulative analysis of attention and inattention, the differences in the frames between the various categories have been calculated. The focus was mainly directed towards the road, which recorded 78% of frames, as users tend to focus on both the pavement close to their location point and ahead so that they can ready and responsive to each vehicle manoeuvre (Table 2) [59]. The greatest number of inattention frames is directed towards the GPS sensor (AVERAGE = 61%) placed in the handlebar with its path monitoring display (Table 3). The GPS is part of the Gamin sensors which show the user the correct route. This could be caused by the fact that the route that cyclists have to follow to move from the point of origin to the point of destination is not well defined, due to the absence of a proper bike lane. However, the fixations directed towards GPS distract from the main task of cycling; that is, from everything that includes the road, at the level of infrastructure and signage, to the traffic and the users that compose it.

**Table 3.** Categories of inattention.

| Categories | Total Frames | Total Fixation Duration [s] | Average Percentage [%] |
|---|---|---|---|
| Buildings | 3066 | 123 | 16 |
| Vegetation | 774 | 31 | 4 |
| Street lamps | 601 | 25 | 3 |
| Sky | 7 | 0.28 | 0 |
| Handlebar | 2136 | 85 | 11 |
| Pedals | 721 | 29 | 4 |
| GPS | 11,410 | 456 | 61 |

During the test, the interaction with crossing pedestrians was also evaluated in Zone 2. Looking at the cyclists' behaviour, 48% did not stop, instead doing quite the opposite by increasing their speed to avoid the pedestrian, without paying particular attention to them. Only 2 out of 20 users look at these weak road users and modulate their cycling behaviour in order to give the right-of-way. This behaviour denotes a trend that does not respect the rules of the road and highlights a dangerous attitude. A possible solution could be to regularize the pedestrian crossing through a traffic light, which imposes a stop not only for bicycles but for all vehicle categories. Zone 2 also has a large number of traffic lights along the route. Nevertheless, participants did not pay much attention to intersections, recording only 4% of frames for the traffic lights decreasing to 2% for the pedestrian passage. This important result underlines the first critical point of the infrastructure that does not allow the cyclists' attention to be focused on intersections, also finding that as many as 11% of these passed the crossing while the traffic light was red [31].

*3.2. Simulated Test*

The test outputs in the bicycle simulator (Figure 7) report a high attention trend with an average fixation duration of 8.28 min (SD:0.02) over 10.33 min of the entire route (Table 4). Considering the different areas, there is a weighted average of attention equal to 85% in the 1A zone, 87% in zone 2, 81% in zone 3, and finally 83% in zone 1B. The two zones with the lowest number of attention frames are the last two zones, when the participants, after familiarizing themself with the simulator, put their attention toward objects that catch their curiosity, such as buildings and vegetation. The greatest inattention occurs in zone 3, where cyclists interact with pedestrians, not with vehicles; this feature shows a lower workload for cyclists, who then can focus on secondary elements. According to the

sickness questionnaire, the lowest percentage of attention in this zone is justified by the rapid collapse of workloads as they pass by a road containing cyclists and cars, to a stretch shared between cyclists and pedestrians. In fact, they do not feel fatigued and do not have vertigo or view and cognitive difficulties, unlike the remaining areas [60]. Although the categories of attention such as pedestrian crossing and traffic lights are poorly focused on by cyclists (AVERAGE = 4%), users have a different subjective perception. According to an objective point of view, these infrastructural elements are not so attractive and underline the loss of their main function, i.e., to decrease the speed and suggest stopping at the dangerous point of alignment. The subjective questionnaires highlight the opposite, that not only are the traffic lights visible but also the cyclists respect the traffic light phases in 87% of cases (Table 5). However, it is important to underline that the objective point of view appears to be that which denotes the actual and most dangerous behaviour of cyclists as they, as seen by the filmmakers, pass the traffic light when it is red.

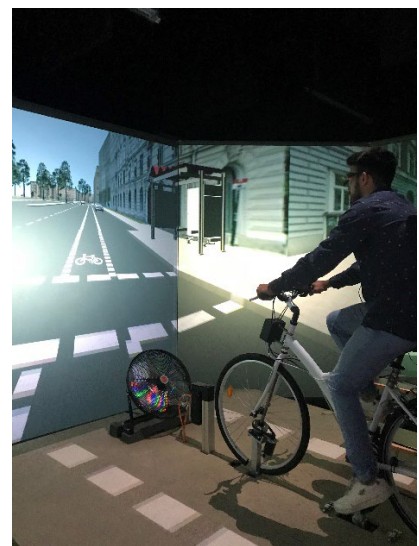
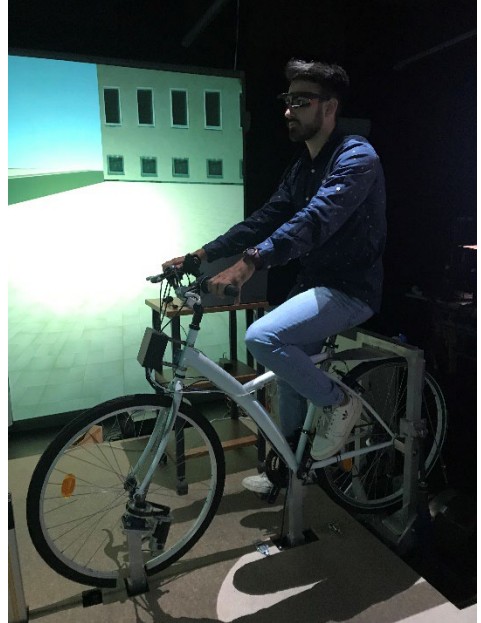
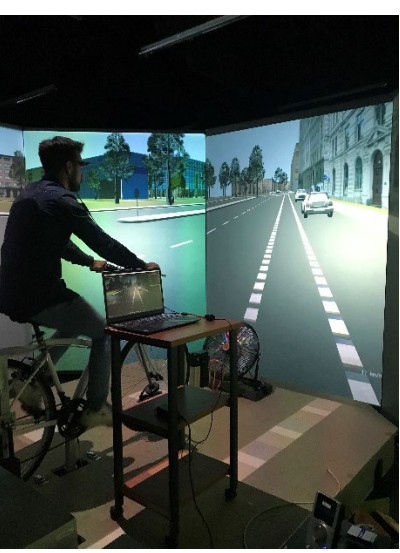

**Figure 7.** Simulated Test.

**Table 4.** Total fixation and duration considering the attention and the inattention.

| Zones | Total Frames | Total Fixation Duration [s] | Fixation Duration of Attention [s] | Percentage of Attention [%] | Fixation Duration of Inattention [s] | Percentage of Inattention [%] |
|---|---|---|---|---|---|---|
| 1A | 12,168 | 487 | 386 | 79 | 101 | 21 |
| 2 | 24,315 | 973 | 813 | 84 | 159.48 | 16 |
| 3 | 16,712 | 668 | 492 | 74 | 176.24 | 26 |
| 1B | 8771 | 351 | 297 | 85 | 53.6 | 15 |

A second example that underlines the difference between objective and subjective perception is linked to a specific interaction in zone 2. In fact, a wheelchair has been programmed to pass over a non-traffic light-controlled pedestrian crossing. The questionnaires show that 85% of users say they have enough time to brake safely to permit the crossing. By contrast, only 20% of cyclists stop to give the right of way. Both examples highlight how cyclists believe they are respecting the rules of the road, while their actual behaviour is the opposite. This important result also denotes the effectiveness of the Pupil Core as a tool that can detect the attention of cyclists, as it is in line with the attitude that is recorded by the videos.

**Table 5.** Categories of attention.

| Categories | Total Frames | Total Fixation Duration [s] | Average Percentage [%] |
|---|---|---|---|
| Sidewalk | 19,203 | 768 | 9 |
| Street | 159,839 | 6394 | 78 |
| Car | 9355 | 374 | 5 |
| Parked car | 4053 | 162 | 2 |
| Pedestrian | 3272 | 131 | 2 |
| Bicycle | 2026 | 81 | 1 |
| Horizontal Signs | 0 | 0 | 0 |
| Vertical Signs | 25 | 1 | 0 |
| Pedestrian passage | 2782 | 111 | 1 |
| Traffic light | 5476 | 219 | 3 |

The categories of inattention deal with 64% of frames dedicated to buildings and 31% to vegetation, such as trees, bushes, and meadows (Table 6). Such distraction, localized in particular near the crossings (average = 61%), confirm the ineffectiveness of such infrastructural elements. In the analysis of inattention, it has been possible to identify cycling behaviour that follows indices in contrast to average performance. User 8, in fact, has a higher percentage of frames focused on elements of inattention, about 57%. In particular, the user registers twice as many frames facing buildings as the street; in the same way, they observed the vegetation for a much longer time than the sidewalks. This objective evaluation is opposed to the perception of the user themself. In the questionnaire, they stated that they pay attention to the road, having a clear path to follow, and in particular the intersections. Additionally, in this case there is a gap between objective and subjective perception that underlines a different point of view also for the cognitive load that highlights a behaviour far from the rules of the road.

**Table 6.** Categories of inattention.

| Categories | Total Frames | Total Fixation Duration [s] | Average Percentage [%] |
|---|---|---|---|
| Buildings | 23,795 | 952 | 64 |
| Vegetation | 11,535 | 461 | 31 |
| Street lamps | 45 | 2 | 0 |
| Sky | 1106 | 44 | 3 |
| Handlebar | 744 | 30 | 2 |
| Pedals | 17 | 0.68 | 0 |
| GPS | 0 | 0 | 0 |

In order to define this dichotomy between objective and subjective perception, the NASA TLX questionnaire was administered both before and after the test (Figure 8). Although the average workload values considering a scale from 1 (low) to 20 (high) are low, it is good to highlight that the expectation of mental commitment is higher (AVERAGE = 4; SD = 0.59) than the real, recorded after the test (AVERAGE = 3.8; SD = 0.91). As a result, participants can define the task of cycling as more challenging concerning their perception than reality, as they are calmer and more satisfied with their performance. Therefore, there is a perception which is opposite to the objective evaluation carried out through a frame-by-frame analysis, where the percentage of attention is high, which goes to confirm the subjective results of the test. If the before conditions are compared with the after conditions, moreover, a decrease in mental demand and temporal demand is noted, so as to highlight a simplification of the level of difficulty cycling accompanied by rhythms of variation in the perception of the increasingly minor scenarios. The effort also turns out to be less after users have done the testing because they have perceived an increase in the clarity of the path.

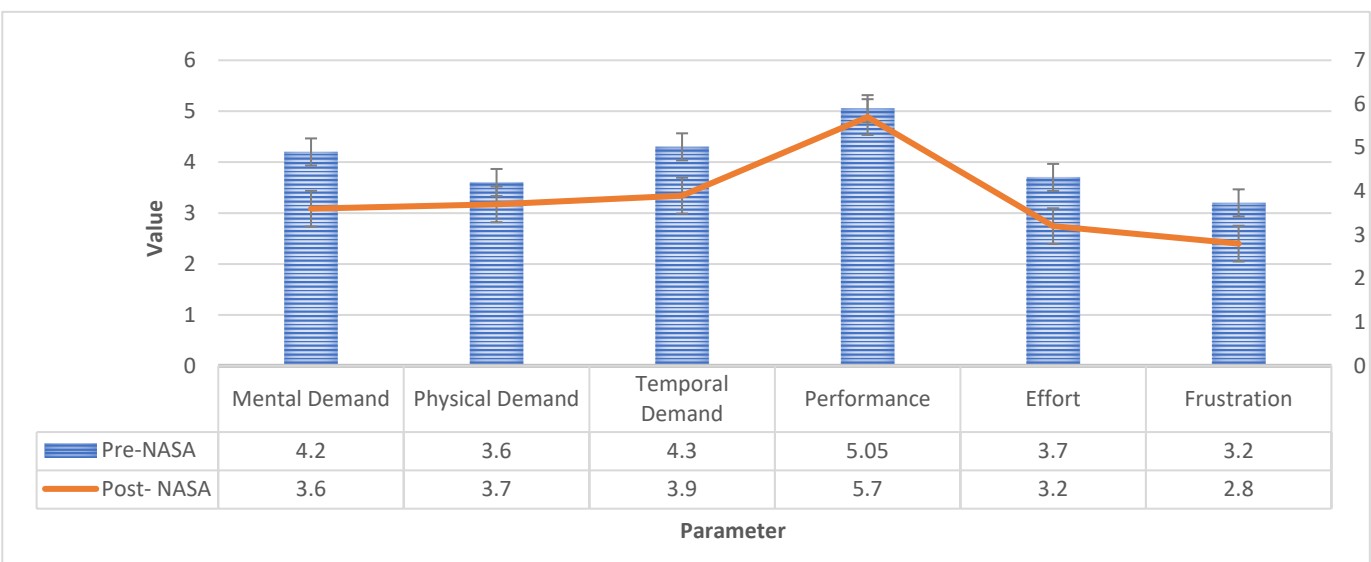

**Figure 8.** Outcomes of the NASA questionnaire comparing the values before and after the experimentation.

*3.3. Comparison*

Comparing the data extrapolated in the site and the simulator, it can be seen that the percentages of attention and inattention of every zone are not the same (Figure 9). In the on-site experiment, the attention rate decreased from the beginning to the end of the circuit, with a maximum value in the first zone (90%). This trend, which highlights not only an adaptation to the route but also a progressive increase in fatigue, is in contrast to the test data in the bicycle simulator. The results show an oscillation of the degree of attention, which is significant, considering the use of the various areas. In fact, the highest values of attention are present in zone 2, where the user must interface with pedestrians, cyclists, vehicles, and buses. They, therefore, appear to be more focused on the primary task, having a greater workload dictated by concentration on their manoeuvres and those that could be accomplished by other road users.

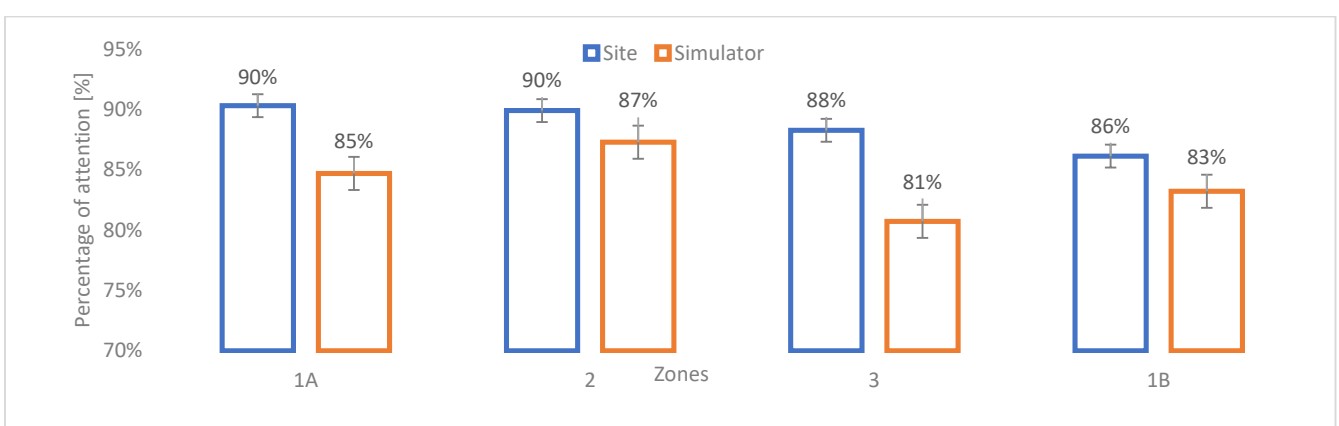

**Figure 9.** Comparison between the percentage of attention for each zone.

By performing a cumulative analysis of the attention data, a 4% difference between the site and simulator ($p < 0.03$) is observed. This data differs from the bibliography [61]. It is expected, in fact, that users, in a closed space such as that of the simulator, are less distracted as they do not suffer from the boundary conditions present in the real scenario. This represents an ulterior important element in the evaluation of the same effectiveness of the tests, emphasizing that the simulation succeeds in reproducing faithfully the real

scenario. Users, in fact, faithfully reflect the behaviours that they would have in realistic cycling on the road according to attention and workload. This factor is further confirmed by the questionnaires, where 90% of cyclists believe that the simulated scenario and the bicycle itself allow them to feel as if they are moving in reality [62]. Moreover, 80% of the participants, being enclosed by seven screens that provide a wide field of view (FOV) and a lateral and rear view, underline that the graphic fluidity (with FPS never below 60) and the feeling of speed are perceived as the real one. Nonetheless, also in this case, it is possible to emphasize a discrepancy between objective evaluation and subjective perception, as the participants unconsciously perceive a higher level of safety within the laboratory which makes them more inclined to lose attention.

## 4. Conclusions

The proposed framework deals with the visual behaviour of cyclists to consider useful insights into objective factors in evaluating their ride style [59,63]. The experiments consist of one road test which includes different cycle tracks (combined cycle and vehicle routes, with or without specific separation signals, pedestrian, and cycle paths) and one simulated experiment. The campaign involved 40 participants who were equipped with a highly innovative tool, the Pupil Core. This eye-tracker allowed a video to be recorded characterized by a circle that focuses on the point of view of each user. The analytical approach uses the attribution and quantification of every single frame to a category such as infrastructure, users, signs, background, or bicycle test. By defining the macro-categories of attention and inattention, it was also possible to quantify the trend for both experiments and then compare them.

First of all, the on-site test showed a low level of inattention, especially towards the subcategory of GPS, which is useful to keep track of the path to follow, but is very often unclear to users. Pedestrian crossings are assessed as the main critical points of the infrastructure. Cyclists do not see them either when assessing the right of way, i.e., in bike crossings as they do not look at traffic lights while crossing, or when they should give priority to a pedestrian crossing. The test in the bicycle simulator, on the other hand, shows an index of inattention related to buildings, as users feel particularly attracted by this simulated environment full of real details. In this test, the on-site assessment of crossings is further confirmed by the simulation of a wheelchair crossing. As many as 80% of users do not give the wheelchair precedence but increase its speed to overtake or completely ignore it. The comparison of the two tests reveals two important common aspects: the high proportion of attention paid to the road and the definition of critical elements of the infrastructure [64]. The first confirms the high road safety throughout the entire route as the elements of the infrastructure allow the cyclist to concentrate on their driving task, confirming the effectiveness of the instrument. In fact, the Pupil Core allows for the evaluation of an objective point of view which is also confirmed by the real behavior of the cyclist, recorded on video. The second aspect, however, makes it possible to identify crossings as places where there is a greater risk of accidents. The factor that most underlines the risk is the low perception of this critical point by users. In fact, only 20% of users approach the crossing by slowing down to give the right way, while 80% say they have a correct behavior approaching this infrastructure element. Furthermore, it is important to underline that the objective perception, resulting from the processing of the Pupil Core data and the videos, is much more realistic as it includes the cyclist's actual attention and mental load.

In most cases, the behaviors of the simulation participants coincided with the behavior of the real scenario participants. Thus, the higher percentage of positive behaviors is attributed to the real scenario, most likely because there are pedestrians and other road users, so the participants pay more attention when performing the test. A major difference between the real scenario test and the simulated scenario test is how the user perceives the road conditions. Although the vibration effects of the road surface were reproduced in the bicycle simulator, it was not possible to represent with the simulator the critical points

which could compromise the cyclist's safety and loss of stability. The most common critical points are the presence of potholes and the irregular surface of the road [60]. Despite this limitation, simulation participants expressed a high level of enjoyment of the test, saying they did not perceive much difference from reality. It is precisely the factors in common between the tests that emphasize the validity of the use of the bicycle simulator. Indeed, the simulator is as close as possible to the real scenario, obtaining objective results very similar to each other, providing visual sensations, vibration movement, and noise. It was possible to compare the two scenarios by studying the visual behavior of test participants, and this represents the innovative aspect of this research.

The comparison between the objective perception, given by the analysis of the instruments, and the subjective one, linked to the administration of the questionnaires, denotes how cyclists are unaware of their workload and how demanding the task of cycling is. Finally, the comparison between the two different tests confirmed the highest percentage of attention was towards the driving scene, making cyclists reactive for each maneuver on the road and ensuring a good level of safety for them and other road users.

In future work, the authors aim to increase the capabilities of the simulator to consider also the kinematic parameters of the different scenarios.

**Author Contributions:** Conceptualization, E.M.A. and M.S.; methodology, E.M.A. and V.V.; software, M.S. and H.I.; resources, V.V. and C.L.; data curation, E.M.A. and C.B.; writing—original draft preparation, E.M.A. and C.B.; writing—review and editing, V.V., M.S., H.I. and C.L.; supervision, V.V., H.I. and C.L. All authors have read and agreed to the published version of the manuscript.

**Funding:** This research received no external funding.

**Acknowledgments:** This work was supported by the University of Bologna and the Gustave Eiffel University of Paris.

**Conflicts of Interest:** The authors declare no conflict of interest.

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
