# Peer review of "The Visual Behaviour of the Cyclist: Comparison between Simulated and Real Scenarios"

_infrastructures, doi:10.3390/infrastructures8050092_

Round 1
Reviewer 1 Report
The manuscript entitled “The Visual Behaviour of the Cyclist: The Comparison Between Simulated and Real Scenario” proposes an experimental study on infrastructure critical points and the level of distraction they cause. The paper is interesting, however, these are some topics that I believe could be improved.
MAJOR COMMENTS:
1) INTRODUCTION: In my opinion, the Introduction section should be a concise text that provides a fast initial understanding of the research. For this purpose, it must only consist of a brief introduction to the topic, delimit the research gap, the objective, highlight the novelty of the research, and provide a brief explanation of the methodology to be used. Therefore, I suggest that the Intro section be restructured, and the literature review be presented in the next chapter.
2) LITERATURE REVIEW: All information from previous works that aim to promote a theoretical background to enable the understanding of this study must be placed in a literature review section.
3) METHOD: Item 2.1 - This topic needs to be expanded. The research results depend directly on the evaluation of the forty participants, whose age, cycling experience, prior knowledge of the path, and other characteristics affect their judgment. Therefore, the information provided is not enough to characterize the sample. I suggest providing more demographic data about the participants.
4) METHOD: Item 2.1 – In my opinion, it would be interesting to publish the model of the questionnaires answered by the participants, even if in the form of supplementary files.
5) DISCUSSION AND RESULTS: Although the title of Section 3 is “Discussion and Results”, in my opinion, only the research results have been addressed. In other words, the discussion of these results was not satisfactory. Therefore, I suggest that this section be greatly improved. Authors should conduct a more in-depth discussion on the results obtained, in addition to reflecting on the methodology adopted.
6) CONCLUSIONS: This section needs improvement. Authors must dig a little deeper into the practical implications of the research. In addition, as it is an experimental procedure, it is subjected to several limitations, which must be clearly indicated. Finally, suggestions for future work should be presented.
7) REFERENCES: Although the paper has an extensive bibliography, most of the referenced studies are old. Of the 66 cited papers, only 23 were published in the last 5 years. Therefore, I suggest that the bibliography be updated, so that authors can provide the state of the art on the subject. As indicated by the authors, the use of bicycles in cities has been intensifying in recent years, which has provided a fertile field for research in this field of knowledge.
MINOR COMMENTS:
8) Although it is not within the scope of the review task, I would like to suggest that authors observe the journal's “Instructions for Authors” in order to prepare the manuscript according to its guidelines. Also, I suggest that the general formatting and spelling of the text be revised.
9) I suggest that the figures are better positioned in the text. The impression is that the manuscript has not been properly proofread and needs to be professionally formatted.
10) I suggest that values are not presented in this way “2/4%”, as they may favor ambiguity in their interpretation.
11) I suggest that the graphics in Figures 7 and 8 be better worked on, making them more attractive to the reader.
12) Please avoid repetitive use of the term “In fact”.
Reviewer 2 Report
This paper must first be extensively revised and edited for English grammar and usage.
Major comments:
Please highlight the significance and novelty of this research in the abstract and conclusions. Are there any practical implications of the research (driver or cyclist training, infrastructure design, GPS design)?
This is mainly a paper presenting descriptive statistics. What analyses were conducted in the comparison between the on-road study and the simulator study?
The literature review focuses almost entirely on vehicle drivers, not cyclists. You do not make the connection between the two categories. I suggest you greatly shorten the literature review and the list of references, and add more cyclist discussion and references. I would like to see a discussion of various road use vulnerabilities ranging from pedestrians (most vulnerable) to heavy trucks (least vulnerable) to show the reader where cyclists fit into this framework.
Please provide more detail about the participants. What was the gender breakdown? How about cycling experience, cycling frequency, and route familiarity (which would definitely affect gaze and glance patterns)? Was the simulator experiment conducted in the same city as the on road experiment? If it was, mention why you chose to conduct a between subjects research design versus a within subjects design.
Tables 1 and 4, please add columns for percentages.
Top of page 2, please provide citations for how the glasses provide insight into brain function.
It is the NASA Task Load Index (not task low). You mention a disease questionnaire and later an illness questionnaire, without ever saying what these are. Are you referring to simulator sickness?
Could high fixation on the GPS be indicative of low familiarity with the route?
First paragraph below figure 6. You discuss the ineffectiveness of infrastructure elements. How would such elements ever be effective?
The following paragraph discusses User 8. I would suspect that the calibration failed during the course of the test to produce such an outlier. Did you re-check calibration at the end of each test?
Please move the images of the simulator into the area where it is described.
There are many minor areas for improvement, but these should be mostly handled once the English grammar and usage are corrected.
Reviewer 3 Report
The paper presents the visual behaviour of the cyclist in simulated and real scenario. It presents a good state of art .
Some comments:
1. in pag. 4 it is written that "the simulator route is half Stockholm route because of technical limitations" but results of test are relative to the entire track (zone 1 A-B, 2 and 3). Please clarify if the simulator route is related or not to the different zone
2. author used NASA TLX questionnaire. Please explain the list of questions;
3. please correct the pointed list at pag 5;
4. please correct the title in pag. 6;
5. in pag.7 authors cited GPS sensor but it is not explained before in the instrument. Please add charateristics of GPS used and its use during the test (which measures was collected and why)
6. please extend the conclusion
Round 2
Reviewer 1 Report
Dear authors,
Thanks for providing a revised version of the manuscript entitled “The Visual Behaviour of the Cyclist: The Comparison Between Simulated and Real Scenario”.
I acknowledge the authors' effort to improve the study based on the reviewers' comments, however, in my opinion, some of my major comments were not adequately addressed.
Therefore, I repeat the topics that, in my opinion, still need to be improved so that the paper is suitable for publication in this journal.
MAJOR COMMENTS:
1) METHOD: Item 2.1 - This topic needs to be expanded. The research results depend directly on the evaluation of the forty participants, whose age, cycling experience, prior knowledge of the path, and other characteristics affect their judgment. Therefore, the information provided is not enough to characterize the sample. I suggest providing more demographic data about the participants.
Comments on the author’s 1st review: The data provided are still very limited, given the relevance of the participants to the study. How was the selection of participants conducted? Is 40 (20/20) a significant sample for the study? Are the participants cyclists? That is, are they used to riding their bikes on the street? Please provide more information to enhance the research.
2) DISCUSSION AND RESULTS: Although the title of Section 3 is “Discussion and Results”, in my opinion, only the research results have been addressed. In other words, the discussion of these results was not satisfactory. Therefore, I suggest that this section be greatly improved. Authors should conduct a more in-depth discussion on the results obtained, in addition to reflecting on the methodology adopted.
Comments on the author’s 1st review: Splitting the section into a Results section and a Discussion section does not solve the issue. According to my original comment, in my opinion, the discussion should be deepened, which is not feasible with just the inclusion of a sentence. Author’s must include their perceptions about the study, as well as defend the paper's relevance to the research field.
3) CONCLUSIONS: This section needs improvement. Authors must dig a little deeper into the practical implications of the research. In addition, as it is an experimental procedure, it is subjected to several limitations, which must be clearly indicated. Finally, suggestions for future work should be presented.
Comments on the author’s 1st review: The original comment was about the need to deepen the practical implications of the study. However, the authors decided to exclude the section, which is not pertinent. Please come back with Conclusions section and improve its content.
Author Response
2. METHOD: Item 2.1 - This topic needs to be expanded. The research results depend directly on the evaluation of the forty participants, whose age, cycling experience, prior knowledge of the path, and other characteristics affect their judgment. Therefore, the information provided is not enough to characterize the sample. I suggest providing more demographic data about the participants. Comments on the author’s 1st review: The data provided are still very limited, given the relevance of the participants to the study. How was the selection of participants conducted? Is 40 (20/20) a significant sample for the study? Are the participants cyclists? That is, are they used to riding their bikes on the street? Please provide more information to enhance the research.
According to the review, Item 2.1 has been expanded. In particular, there were added sentences such as: ‘Participants represented a homogeneous and statistically significant sample, considering 22 males (11 for the on-site test, 11 for the simulator test) and 18 females (9 for the on-site test, 9 for the simulator test). They were recruited through social networks and posters in the universities where the tests were carried out. Participants had an average cycling experience of 30.9 years (SD: ±15.9) for males and 26 years (SD: ±15.8) for females and they used to ride the bicycle every day, to reach their places of work or study. For the on-site test, 64% of the users had familiarity with the experiment route (57% of males, 75% of females), instead, no one had used a bicycle simulator [67]. No user knew the purpose of the test, so as not to affect the results. Before the test, all users were provided with relevant information material: the route to be followed, the experimental procedure, and the instruments used, in terms of operation and calibration phase’
2. DISCUSSION AND RESULTS: Although the title of Section 3 is “Discussion and Results”, in my opinion, only the research results have been addressed. In other words, the discussion of these results was not satisfactory. Therefore, I suggest that this section be greatly improved. Authors should conduct a more in-depth discussion on the results obtained, in addition to reflecting on the methodology adopted.
Comments on the author’s 1st review: Splitting the section into a Results section and a Discussion section does not solve the issue. According to my original comment, in my opinion, the discussion should be deepened, which is not feasible with just the inclusion of a sentence. Author’s must include their perceptions about the study, as well as defend the paper's relevance to the research field.
According to the review’s comment, the section ‘Discussion and results’ has been improved.
3. CONCLUSIONS: This section needs improvement. Authors must dig a little deeper into the practical implications of the research. In addition, as it is an experimental procedure, it is subjected to several limitations, which must be clearly indicated. Finally, suggestions for future work should be presented.
Comments on the author’s 1st review: The original comment was about the need to deepen the practical implications of the study. However, the authors decided to exclude the section, which is not pertinent. Please come back with Conclusions section and improve its content.
According to the review’s comment, the section ‘Conclusions’ has been improved.
Reviewer 2 Report
Somehow the first and most important revision request was ignored. "This paper must first be extensively revised and edited for English grammar and usage."
The paper will not be ready for publication until this is complete.
Some revision requests were ignored and not addressed in the author's response. Please go back to the original review and ensure that all issues are addressed or corrected.
The tables (2 and 4) where percentage columns were requested are still not adequate. Please provide the percentage of inattention for each zone (that is much more important than how much data was collected in each zone).
Author Response
Following the auditor’s comments, all 1 revisions were reviewed. In addition, English has been revised according to the correct grammatical rules. Thank you very much for your work.
The detailed point-to-point reply can be found in the attachments.

Reviewer 3 Report
Thanks to authors that clarify all my suggestions
Author Response
Thank you.
Round 3
Reviewer 1 Report
--
Author Response
According to the review, all the paper has been appropriately revised.
Reviewer 2 Report
I have asked for a thorough review of the English grammar and usage and I cannot tell that anything has been done. Here are four examples from the very beginning of the paper.
Line 35: What is adjective analysis? That term in unfamiliar and I cannot find a definition online.
Line 47. The beholder and the sighted is not the way this would be phrased in everyday English (the observer and the target would be much more understandable).
The sentence beginning on line 47 makes no sense: Observe the movements of the gaze and analyze how the individual can reach certain levels of attention and defines visual behavior about specific actions or scenarios determined in the external environment ...
The sentence beginning on line 50 also makes no sense: It allows to record of the opinions
